# Kinetic and Structural Aspects of Glycosaminoglycan–Monkeypox Virus Protein A29 Interactions Using Surface Plasmon Resonance

**DOI:** 10.3390/molecules27185898

**Published:** 2022-09-11

**Authors:** Deling Shi, Peng He, Yuefan Song, Shuihong Cheng, Robert J. Linhardt, Jonathan S. Dordick, Lianli Chi, Fuming Zhang

**Affiliations:** 1National Glycoengineering Research Center, Shandong University, Qingdao 266237, China; 2Center for Biotechnology and Interdisciplinary Studies, Rensselaer Polytechnic Institute, Troy, NY 12180, USA

**Keywords:** A29, chondroitin sulfate, dermatan sulfate, heparin, monkeypox virus, surface plasmon resonance

## Abstract

Monkeypox virus (MPXV), a member of the Orthopoxvirus genus, has begun to spread into many countries worldwide. While the prevalence of monkeypox in Central and Western Africa is well-known, the recent rise in the number of cases spread through intimate personal contact, particularly in the United States, poses a grave international threat. Previous studies have shown that cell-surface heparan sulfate (HS) is important for vaccinia virus (VACV) infection, particularly the binding of VACV A27, which appears to mediate the binding of virus to cellular HS. Some other glycosaminoglycans (GAGs) also bind to proteins on Orthopoxviruses. In this study, by using surface plasmon resonance, we demonstrated that MPXV A29 protein (a homolog of VACV A27) binds to GAGs including heparin and chondroitin sulfate/dermatan sulfate. The negative charges on GAGs are important for GAG–MPXV A29 interaction. GAG analogs, pentosan polysulfate and mucopolysaccharide polysulfate, show strong inhibition of MPXV A29–heparin interaction. A detailed understanding on the molecular interactions involved in this disease should accelerate the development of therapeutics and drugs for the treatment of MPXV.

## 1. Introduction

Before April 2022, monkeypox virus (MPXV) infection in humans was rarely reported outside of endemic areas in Africa. Since MPXV cases were first reported in Europe in early May 2022, more than 39,000 confirmed cases and 12 deaths have been reported in at least 94 countries and locations as of 18 August 2022 [1], prompting the World Health Organization (WHO) to declare the monkeypox outbreak “a public health emergency of international concern”. By August 2022, the WHO assesses the global risk as moderate, except in the European region where the risk is high [2].

Monkeypox is a contagious viral disease affecting humans with symptoms similar to smallpox, including rash, fever, muscle pains, and respiratory symptoms, typically lasting 2–4 weeks. Monkeypox is caused by MPXV, a zoonotic virus [3] belonging to the genus Orthopoxvirus of the family Poxviridae [4]. Orthopoxviruses are double-stranded DNA viruses, of which four species are known to cause disease in humans: vaccinia virus (VACV), cowpox virus (CPXV), variola virus (VARV), and monkeypox virus (MPXV) [5]. MPXVs was initially identified in non-human primate rash lesions in 1958 [6] and was first identified in humans in 1970 [7]. The size of the viral genome is approximately 197 kbp, and it encodes more than 190 open reading frames (ORFs) [8]. The virus replicates in the cytoplasm of infected cells, and its life cycle is shown in Figure 1. MPXV produces two infectious viral particles during replication: intracellular mature virus (MV) and extracellular enveloped virus (EV). MV is released upon cell lysis and is mainly responsible for transmission between animals [9].

Two distinct clades have been identified: the West African clade and the Central African clade. The case–fatality rate in Central Africa in the 1980s was about 10% in non-vaccinated individuals, while there were no fatalities in cases occurring in West Africa [10]. While MPVX generally does not show the large number mutations as seen in RNA viruses, such as SARS-CoV-2, isolates from the 2022 outbreak shared 40 mutations, well above the virus’ standard mutation rate [9]. Monkeypox is clearly less severe than smallpox, with lower mortality (recent case fatality ratio 3%–6%) compared with smallpox (30%) [11]. The genome of the MPXV shows over 96% identity to the VARV. Smallpox vaccination has been reported to provide 85% protection against MPXV [12]. Nevertheless, many mutations in gene sequences of MPXV are alarming to scientists, and effective vaccination or antiviral drugs against MPXV are needed to prevent the spread.

Orthopoxviruses are highly homologous at the DNA and protein levels. Several virion proteins have been shown to be crucial for the binding of the virion to the cell surface. For example, the initial association of MV with the cell is thought to occur through the binding of ubiquitously expressed glycosaminoglycans (GAGs) to the A27L [13,14,15], D8L [16], and H3L [17] proteins. In addition to VACV, other poxviruses such as CPXV, rabbitpox virus, Shope fibroma virus, and myxoma virus also bind to heparan sulfate (HS) [18]. VACV A27 consists of 110 amino acid residues that can be divided into four functional domains: an N-terminal signal peptide; a Lys/Arg-rich domain known as the heparin binding site (HBS); an α-helical coiled-coil domain; and a C-terminal leucine zipper motif. The HBS sequence of VACV A27 is “STKAAKKPEAKR”, while the sequence in MPXV A29 is “STKAAKNPETKR”. MPXV A29 (a homolog of VACV Copenhagen A27) binds to heparin with similar affinity, as with VACV A27, regardless of the sequence changes in the HBS [19]. Antibodies against L1R protein (an outer membrane protein of the MV) can neutralize viral infectivity, suggesting that L1R may also play a role in viral particle entry [20]. Apart from the structure/function correlation, the specific GAG-binding mechanism of MPXV A29 remains unclear. Whether L1R can bind to GAGs also needs to be confirmed.

GAGs, commonly expressed in most cell types, are a family of negatively charged linear polysaccharides, including heparin/HS, chondroitin sulfate (CS)/dermatan sulfate (DS), hyaluronan (HA), and keratan sulfate (KS). The chemical structures of GAGs have been well-characterized and are typically comprised of repeating disaccharides of a hexuronic acid and an *N*-acetyl hexosamine, modified with sulfo monoester groups (Figure 2). GAGs interact with a wide range of proteins to mediate many pathological processes/diseases, such as infectious disease [21], cardiovascular disease, inflammation, and cancer [22]. Elucidation of the mechanism by which GAGs interact with proteins is essential for the discovery of novel therapeutics.

In this study, we explore the role of GAGs, particularly heparin and heparin-derived oligosaccharides, in MPXV infection. A small library of sulfated glycans and highly negatively charged compounds, including heparin oligosaccharides, desulfated heparins, pentosan polysulfate (PPS), and mucopolysaccharide polysulfate (MPS), was assembled and evaluated for binding to MPXV proteins. Surface plasmon resonance (SPR) was used to provide direct quantitative analysis of the label-free molecular interactions in real-time. Our results suggest that MPXV A29 binds to heparin, DS, and CS, whereas L1R has no affinity for heparin. Furthermore, PPS and MPS showed the strongest inhibition of interaction between heparin and MPXV A29.

## 2. Results and Discussion

### 2.1. Binding Affinity and Kinetics Measurement of MPXV Protein–GAG Interactions

The L1R protein is encoded by the *L1r* gene and is highly conserved among the Orthopoxviruses. L1R is a myristylated 23–29 kDa membrane protein located on the surface of MVs and beneath the envelope on EVs. The structure of L1R has been solved, revealing a molecule comprised of a bundle of α-helices packed against a pair of two-stranded β-sheets, held together by four loops [23]. An SPR heparin chip was prepared to quantify the binding of L1R. The binding signal of MPXV L1R–heparin interactions at different concentrations are shown in Table 1. Although L1R can neutralize viral infectivity and play a role in particle entry, SPR results showed there was essentially no binding between MPXV L1R protein and heparin. The bindings of L1R to other GAGs (DS, CSA, and CSE) were also negligible, and the signals were not concentration-dependent (data not shown).

Heparin and HS interactions with VARV A27 and MPXV A29 have been demonstrated by other researchers [19]. It is unknown whether other GAGs have broad binding affinity for MPXV A29, so the strength of their binding remains to be determined. Heparin/HS are GAGs, with HS produced by all cell types, which is a component of the extracellular matrix [24]. Heparin is distinct from HS (less than one sulfo group per disaccharide unit), in that it is produced primarily by mast cells, with a higher degree of sulfation (approximately three sulfo groups per disaccharide unit). DS, also previously referred to as CSB, is found primarily in skin, as well as in blood vessels, heart valves, tendons, and lungs. DS consists of repeating disaccharide units, *N*-acetyl galactosamine (GalNAc) and iduronic acid (IdoA), which are sulfated at multiple positions and have, on average, one sulfo group per disaccharide unit [25]. CS is an important structural component of cartilage. CS chains are unbranched polysaccharides of variable lengths, containing two alternating monosaccharides: glucuronic acid (GlcA) and GalNAc. CS can be divided into CSA (chondroitin-4-sulfate) and CSC (chondroitin-6-sulfate), both having, on average, one sulfo group per disaccharide unit; and CSD (chondroitin-2,6-sulfate) and CSE (chondroitin-4,6-sulfate), both having, on average, two sulfo groups per disaccharide unit. Here, different GAG chips, including heparin, DS, CSA, and CSE chips, were prepared to quantify the binding affinity for MPXV A29. Sensorgrams of the interactions of GAGs with MPXV A29 are shown in Figure 3.

The resulting sensorgrams were used to determine binding kinetics and affinity (i.e., association rate constant: *k_a_*; dissociation rate constant: *k_d_*; and binding equilibrium dissociation constant: K_D_, where K_D_ = *k_d_/k_a_*) by globally fitting the entire association and dissociation phases using a 1:1 Langmuir-binding model (Table 2). The binding affinity between heparin and MPXV A29 is significantly higher than between heparin and MPXV L1R. Other GAGs, such as DS, CSA, and CSE, also bind to MPXV A29. A comparison of the binding kinetics and affinities showed that GAGs with a higher degree of sulfation exhibit stronger binding affinity to MPXV A29, suggesting that binding is influenced by the level of sulfation within the GAG.

### 2.2. SPR Solution Competition between Surface-Immobilized Heparin and Heparin Oligosaccharides and Desulfated Heparins

Heparin and HS are comprised of linear chains of repeating disaccharide units, consisting of a glucosamine and uronic acid. The initial disaccharide unit that constitutes the growing chain during biosynthesis has a D-glucuronic acid *β*1 → 4 linked to a D-*N*-acetylglucosamine. These units are linked to each other by an *α*1 → 4 linkage. The subsequent modifications proceed in a sequential manner, beginning with the *N*-deacetylation and *N*-sulfation of glucosamine residues within the chains. This is followed by epimerization of the GlcA to IdoA and *O*-sulfation at the C-2 of the uronic acid and the C-6 of the glucosamine. The final modification step in this pathway is *O*-sulfonation at the C-3 of the glucosamine [26]. Both heparin and HS chains are polydisperse, with a broad molecular weight distribution. HS chains are generally longer than heparin chains and have an average molecular weight of ~30 kDa, compared to ~15 kDa for heparin.

We prepared oligosaccharides of different lengths, from tetrasaccharide (dp4) to octadecasaccharide (dp18), by enzymatic degradation of heparin, to examine the effect of the saccharide chain length of heparin on the MPXV A29 interaction. Solution/surface competition experiments were performed using SPR to examine the inhibition of different heparin oligosaccharides for the interaction between heparin (on the surface) with MPXV A29. The same concentration (1000 nM) of heparin oligosaccharides was mixed in the MPXV A29 protein (250 nM)/heparin interaction solution. Solution competition studies between heparin and heparin oligosaccharides are shown in Figure 4A,B. All the oligosaccharides of heparin showed inhibition of MPXV A29 binding to the heparin surface, compared with the heparin control. Heparin inhibited the binding of MPXVA29 to the surface-immobilized heparin by 30%. Heparin oligosaccharides from dp4 to dp18 inhibited 15% to 27% of the binding, but there was no apparent glycan-length binding dependence.

Furthermore, to address the chemical structure leading to heparin competition, we obtained different heparin analogs by chemical modification that contained reduced sulfate content, while having approximately the same molecular weight (and chain length). Solution/surface competition experiments were also performed using SPR, to examine the inhibition of MPXV A29–heparin interactions by different desulfated heparins. Solution competition studies between heparin and desulfated heparins are shown in Figure 4C,D. Specifically, 2-desulfated heparin, 6-desulfated heparin, and *N*-desulfated heparin showed weaker inhibition of MPXV A29 binding to the heparin surface, compared with the heparin control. Therefore, removing any sulfate from heparin reduces its binding affinity to MPXV A29 protein. However, the differences among the three desulfated samples were not significant, suggesting that binding was not specific for sulfo group position and was mainly dependent on the presence of sufficient charge.

### 2.3. SPR Solution Competition Study on the Inhibition of Different Heparin Analogs to the Interaction between Surface-Immobilized Heparin with MPXV A29

PPS, a heparin mimetic with a highly sulfated polysaccharide backbone, is synthesized through the chemical sulfonation of a plant-derived β-(1 → 4)-xylan. MPS is a semisynthetic GAG with a backbone that is isolated from mammalian cartilage before its chemical sulfation. Importantly, PPS is an FDA-approved active pharmaceutical ingredient of the oral drug Elmiron. The FDA has approved PPS as an oral anti-thrombotic agent for the management of patients with interstitial cystitis, and it is also used for clinical disorders such as antagonism of enzymatic activities and inhibition of HIV infectivity. MPS has been used for the topical treatment of superficial phlebitis, hematomas, and sports-related injuries [27].

MPXV A29 was premixed with the same concentrations of PPS, MPS, or heparin, before injection into the heparin chip. When the active binding sites on the MPXV A29 were occupied by glycan in solution, its binding to the surface-immobilized heparin decreased, resulting in a reduction in signal. Solution competition between heparin and sulfated glycans is shown in Figure 5. PPS and MPS potently inhibited the MPXV–heparin interaction by 62% and 69%, respectively. This could be due to the level of sulfation being higher for MPS and PPS compared with heparin. The average heparin disaccharide contains ~2.7 sulfo groups, while MPS disaccharide has >4 sulfo groups and PPS disaccharide has >3 sulfo groups; the high level of sulfo groups enable strong interaction with MPXV A29.

## 3. Materials and Methods

### 3.1. Materials

MPXV L1R and MPXV A29 proteins were purchased from Sino Biological Inc. Porcine intestinal heparin, with an average molecular weight of 15 kDa and polydispersity of 1.4, was purchased from Celsus Laboratories (Cincinnati, OH). The *N*-desulfated heparin (14 kDa) and 2-*O*-desulfated IdoA heparin (13 kDa) were prepared according to Yates et al. A 6-*O*-desulfated heparin (13 kDa) was provided by Lianchun Wang from the University of South Florida. The heparin oligosaccharides, including tetrasaccharide (dp4), hexasaccharide (dp6), octasaccharide (dp8), decasaccharide (dp10), dodecasaccharide (dp12), tetradecasaccharide (dp14), hexadecasaccharide (dp16), and octadecasaccharide (dp18), were prepared by controlled partial heparin lyase I treatment of bovine lung heparin (Sigma), followed by size fractionation. The GAGs used were chondroitin sulfate A (20 kDa) from porcine rib cartilage (Sigma, St. Louis, MO, USA), dermatan sulfate (DS; also known as chondroitin sulfate B, 30 kDa, from porcine intestine; Sigma), and chondroitin sulfate E (20 kDa, from squid cartilage; Seikagaku). Pentosan polysulfate (PPS; 6.5 kDa) was from Bene Pharma (Munich, Germany). Mucopolysaccharide polysulfate (MPS; 14.5 kDa) was purchased from Luitpold Pharma (Munich, Germany). Sensor SA chips were from Cytiva (Uppsala, Sweden). SPR measurements were performed on a BIAcore 3000 or T200 SPR (Uppsala, Sweden), operated using Biaevaluation software (version 4.0.1 or 3.2).

### 3.2. Preparation of GAG Biochips

The preparation of biotinylated GAGs was as follows: 2 mg of GAGs (heparin, DS, CSA or CSE) (in 200 µL of water) and 2 mg of amine–PEG3–Biotin (Thermo Scientific, Waltham, MA) were mixed with 10 mg of NaCNBH_3_. The initial reaction was carried at 70 °C for 24 h, then a further 10 mg of NaCNBH_3_ was added, and the reaction continued for another 24 h. After completing the reaction, the mixture was desalted with a spin column (3000 molecular weight cut-off). Biotinylated GAGs were freeze-dried for chip preparation. The biotinylated GAGs were immobilized onto streptavidin (SA) chips, in accordance with the protocol of the manufacturer. In brief, 20 μL solution of the GAG-biotin conjugate (0.1 mg/mL) in HBS-EP+ buffer (0.01 M 4-(2-hydroxyethyl)-1-piperazineethanesulfonic acid, 0.15 M NaCl, and 3 mM ethylenediaminetetraacetic acid, 0.05% surfactant P20, pH 7.4) was injected over flow cell 2 (FC2), 3 (FC3) and 4 (FC4) of the SA chips, at a flow rate of 10 μL/min. The successful immobilization of GAGs was confirmed by the observation of a ~200 resonance unit (RU) increase in the sensor chip. The control flow cell (FC1) was prepared by 1 min injection with saturated biotin.

### 3.3. Binding Kinetics and Affinity Measurement of Interaction between GAGs and MPXV Proteins

MPXV L1R protein and MPXV A29 protein were diluted in HBS-EP buffer. Different dilutions of protein samples were injected at a flow rate of 30 µL/min. At the end of the sample injection, the same buffer was flowed over the sensor surface to facilitate dissociation. After a 3 min dissociation time, the sensor surface was regenerated by injecting with 30 µL of 2 M NaCl. The response was monitored as a function of time (sensorgram) at 25 °C.

### 3.4. Evaluation of the Inhibition Activity of Sulfated Glycans on Heparin–MPXV Protein Using Solution Competition

For testing of inhibition of MPXV protein–heparin interaction, 250 nM of protein was premixed with 1000 nM of different sulfated glyans in HBS-EP+ buffer and injected over the heparin chip at a flow rate of 30 µL/min. At the end of sample injection, the same buffer was flowed over the sensor surface to facilitate dissociation. After dissociation, the sensor surface was regenerated by injecting with 30 µL of 2 M NaCl. The response was monitored as a function of time (sensorgram) at 25 °C. For each set of competition experiments, a control experiment (only protein, without heparin or oligosaccharides) was performed to ensure the surface was completely regenerated and that the results obtained between runs were comparable. When the active binding sites on the proteins were occupied by sulfated glycans in solution, the binding of the proteins to the surface-immobilized heparin decreased, resulting in a reduction in signal in RU.

## 4. Conclusions

SPR analysis confirmed the interactions of GAGs, including heparin, DS, CSA, and CSE, with the MPXV A29 protein. No affinity was observed between heparin and the MPXV L1R protein. We explored the binding affinity of sulfated glycans with different structures to MPXV A29. Solution competition analysis between surface-immobilized heparin with oligo-heparins from dp4 to dp18 showed that the binding was not length-dependent. Compared with heparin, heparin desulfated at different positions showed lower binding affinity, suggesting the binding is associated with the density of negative charges. In a competition SPR assay, PPS and MPS in solution showed remarkable inhibition activity against chip-surface heparin binding to the MPXV. These results suggest that GAG binding to MPXV A29 is dependent on charge interactions. Based on our findings and previous studies, we propose a model on how GAGs may facilitate host-cell entry of MPXV (Figure 6). PPS and MPS have also shown potential use as therapeutic and/or preventative antiviral drugs. Future studies are needed to investigate the structure–activity relationships, bioavailability, and antiviral activity of low-molecular-weight PPS and MPS.

## Figures and Tables

**Figure 1 molecules-27-05898-f001:**
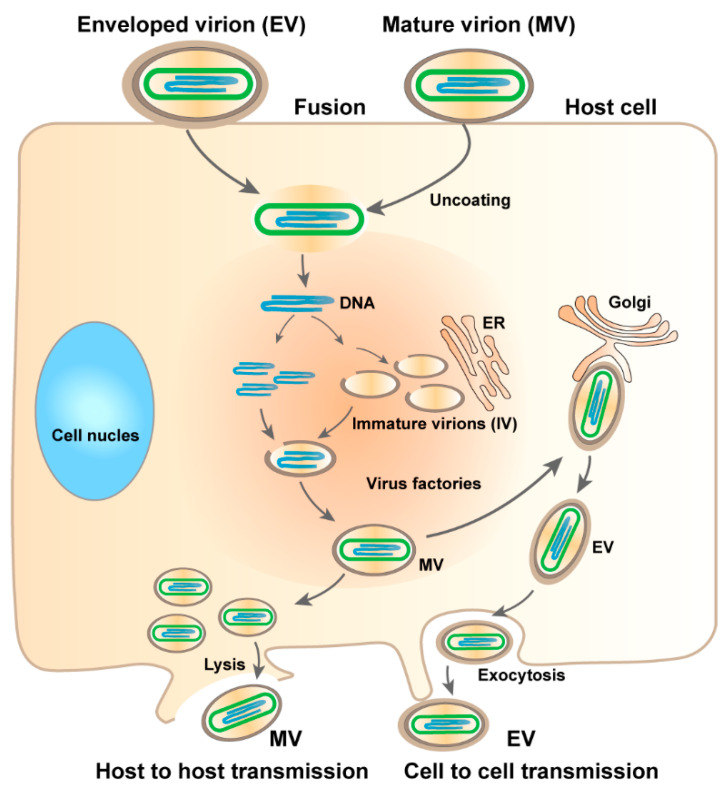
Life cycle of MPXV. EV enters the host cell by fusion and MV by micropinocytosis or fusion. Within the viral factory, immature virions (IVs) are assembled to form MVs. Some MVs are wrapped to form EVs. Virus exits via budding of EVs or by cell lysis to release MVs.

**Figure 2 molecules-27-05898-f002:**
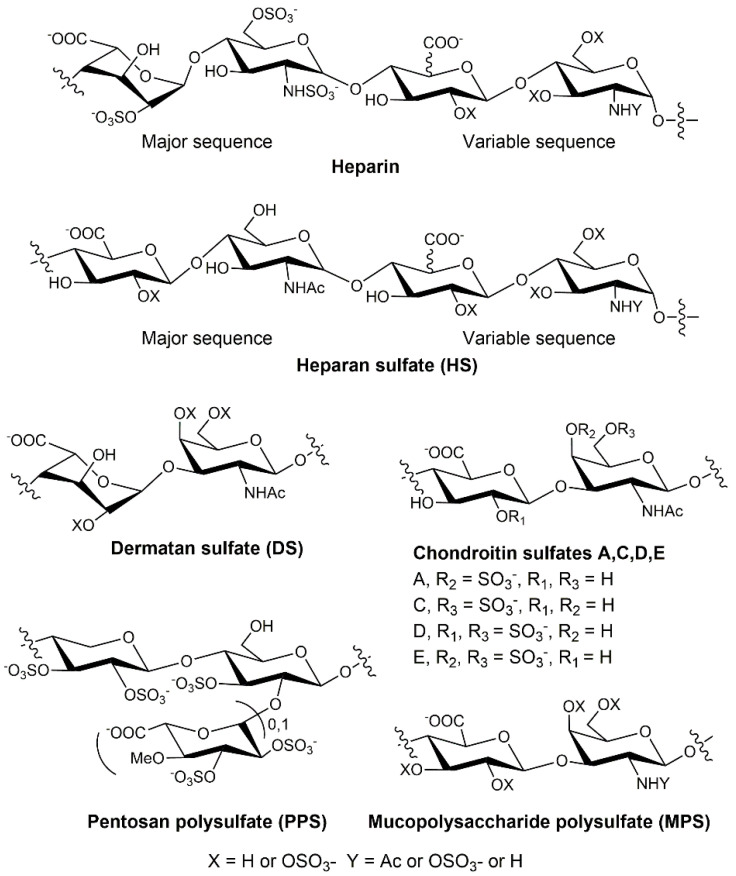
Chemical structures of GAGs (including heparin, HS, CS, and DS) and heparin analogs (PPS and MPS).

**Figure 3 molecules-27-05898-f003:**
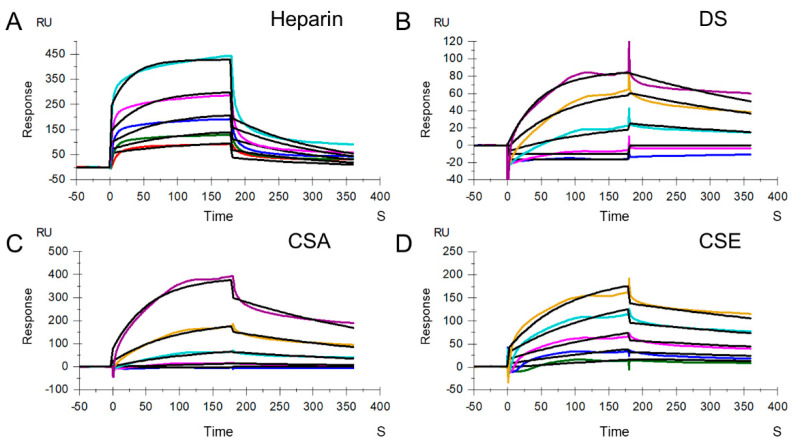
SPR sensorgrams of MPXV A29 binding with GAGs. (**A**) SPR sensorgrams of MPXV A29 binding with heparin. Concentrations of MPXV A29 (from top to bottom) are 1000, 500, 250, 125, and 63 nM, respectively. (**B**) SPR sensorgrams of MPXV A29 binding with DS. Concentrations of MPXV A29 (from top to bottom) are 4000, 2000, 1000, 500, and 250 nM, respectively. (**C**) SPR sensorgrams of MPXV A29 binding with CSA. Concentrations of MPXV A29 (from top to bottom) are 4000, 2000, 1000, 500, and 250 nM, respectively. (**D**) SPR sensorgrams of MPXV A29 binding with CSE. Concentrations of MPXV A29 (from top to bottom) are 2000, 1000, 500, 250, and 125 nM, respectively. The black curves are the fits using models from T200 Evaluate software 3.2.

**Figure 4 molecules-27-05898-f004:**
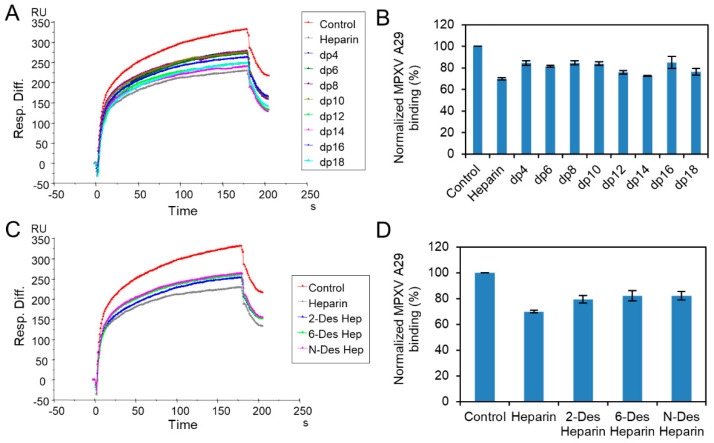
Solution competition between heparin and heparin oligosaccharides/desulfated heparins. (**A**) SPR sensorgrams of MPXV A29–heparin interaction competing with different oligo-heparins. Concentration of MPXV A29 is 250 nM mixed with 1000 nM of different oligo-heparins. (**B**) Bar graphs (based on triplicate experiments with standard deviation) of normalized MPXV A29 binding preference to surface heparin, by competing with different oligo-heparins. (**C**) SPR sensorgrams of MPXV A29–heparin interaction competing with different desulfated heparins. Concentration of MPXV A29 is 250 nM mixed with 1000 nM of different desulfated heparins. (**D**) Bar graphs (based on triplicate experiments with standard deviation) of normalized MPXV A29 binding preference to surface heparin, by competing with different desulfated heparins.

**Figure 5 molecules-27-05898-f005:**
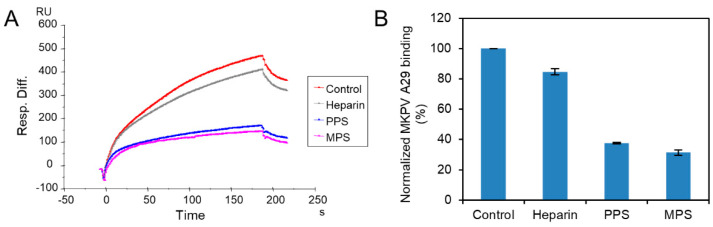
Solution competition between heparin and sulfated glycans. (**A**) SPR sensorgrams of MPXV A29–heparin interaction, by competing with PPS or MPS. Concentration of MPXV A29 is 250 nM mixed with 1000 nM of sulfated glycans. (**B**) Bar graphs (based on triplicate experiments with standard deviation) of normalized MPXV A29 binding preference to surface heparin, by competing with PPS or MPS.

**Figure 6 molecules-27-05898-f006:**
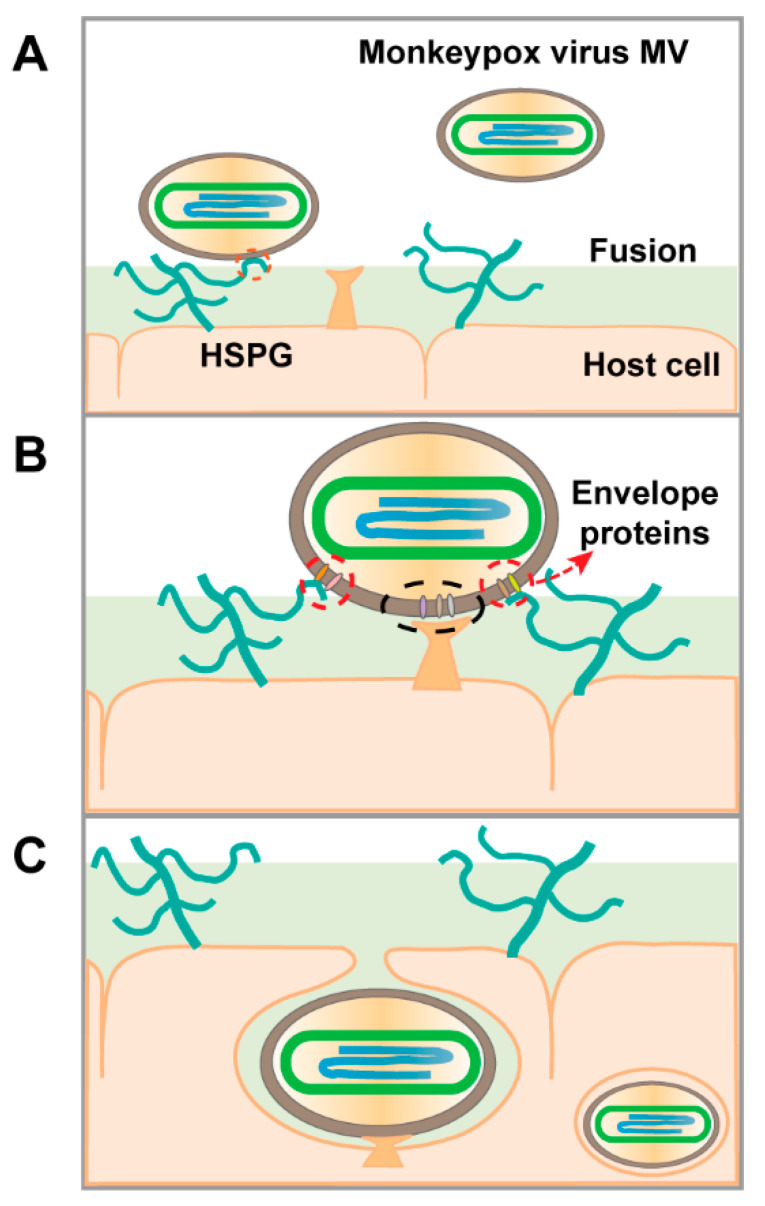
Proposed model of MPXV host-cell entry. (**A**) Virions land on the host-cell surface by binding to heparan sulfate proteoglycan (HSPG). (**B**) Then, host-cell surface proteases initiate viral-host-cell membrane fusion. (**C**) Finally, virions enter the host cell.

**Table 1 molecules-27-05898-t001:** The binding affinity of MPXV L1R–heparin interactions.

MPXV L1R (nM)	Binding Signal (RU) *
500	−4.1 ± 0.4
1000	−1.3 ± 1.0
5000	10.2 ± 1.3

* The data with (±) in parentheses are standard deviations from triplicate experiments.

**Table 2 molecules-27-05898-t002:** Summary of kinetic data of MPXV A29 binding with heparin, DS, CSA, and CSE.

	*k_a_* (M^−1^s^−1^)	*k_d_* (s^−1^)	*K_D_* (M)
Heparin	2.8 × 10^4^ (±280) *	7.1 × 10^−3^ (±5.0 × 10^−5^) *	2.6 × 10^−7^ (±1.7 × 10^−8^) **
DS	4.5 × 10^3^ (±54)	2.8 × 10^−3^ (±2.6 × 10^−5^)	6.2 × 10^−7^ (±1.7 × 10^−9^)
CSA	3.7 × 10^3^ (±120)	3.2 × 10^−3^ (±5.4 × 10^−5^)	8.4 × 10^−7^ (±7.1 × 10^−9^)
CSE	4.9 × 10^3^ (±72)	1.5 × 10^−3^ (±4.7 × 10^−5^)	3.1 × 10^−7^ (±6.1 × 10^−9^)

* The data with (±) in parentheses are the standard deviations (SD) from global fitting of five injections. ** Standard deviation (SD) on triplicate experiments.

## Data Availability

Not applicable.

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
