# Peer review of "Kinetic and Structural Aspects of Glycosaminoglycan–Monkeypox Virus Protein A29 Interactions Using Surface Plasmon Resonance"

_molecules, 2022, doi:10.3390/molecules27185898_

Round 1

Reviewer 1 Report

In this manuscript, the author investigates MPXV A29 (a homolog of VACV A27) interaction with GAGs, the competition interaction with some heparin analogs, and also shows no affinity between heparin (GAG) and MPXV L1R MPXV L1R protein. It provides information to understand how MPXV enters the host cell. The inhibition of different heparin analogs to the interaction between surface-immobilized heparin with MPXV A29 shows novelty and importance.

However, there are some questions about the result of the manuscript.

  1. The author only shows Table 1 of L1R binding with heparin. In Table 1, it shows that at 1 uM and 5 uM the RU already increased. And it can not decide the binding ability of molecules only by this preliminary data and only RU of the binding signal, it is better to show the same binding kinetics and affinity assay. At the same time why the author only shows the result of L1R with Heparin but not other GAGs?
  2. How many repeats of the result Table 3, because this is no deviation data about KD value.
  3. In the 2.2 and 2.3 result part, the conclusion is not solid, especially the conclusion in 2.3, the RU does not mean the binding ability, only the ka or KD value can show the binding ability. Such as, in result part 2.3, the decrease of RU may be because PPS or MPS change the binding mode of A29 to heparin. The author can mix the PPS or MPS with A29, and then use the different concentrations to check the difference of ka or KD value by SPR.
  4. Does PPS or MPS have the unspecific binding with the Biotin labeled the heparin SA chips?
  5. What is the concentration of PPS or MPS pre-mix with A29, in the manuscript the author mentions the same concentration (Page 6, 209), but in the Fig.5 legend, it is “MPXV A29: 250 nM 221 mixed with 1000 nM of sulfated glycans”.

Author Response

Reviewer #1

In this manuscript, the author investigates MPXV A29 (a homolog of VACV A27) interaction with GAGs, the competition interaction with some heparin analogs, and also shows no affinity between heparin (GAG) and MPXV L1R MPXV L1R protein. It provides information to understand how MPXV enters the host cell. The inhibition of different heparin analogs to the interaction between surface-immobilized heparin with MPXV A29 shows novelty and importance.

However, there are some questions about the result of the manuscript.

  1. The author only shows Table 1 of L1R binding with heparin. In Table 1, it shows that at 1 uM and 5 uM the RU already increased. And it can not decide the binding ability of molecules only by this preliminary data and only RU of the binding signal, it is better to show the same binding kinetics and affinity assay. At the same time why the author only shows the result of L1R with Heparin but not other GAGs?

Response: The SPR signals resulting from the binding of L1R to heparin and other GAGs were too weak to provide kinetic results. In addition, the signals were not concentration dependent (see revised Table 1). In our revised manuscript we have changed “binding affinity” to “binding signal”.

Table 1. The binding signal of MPXV L1R-GAG interactions.

500 (nM)

1000 (nM)

5000 (nM)

MPXV L1R-Heparin

-4.1 (± 0.4) RU

-1.3 (± 1.0) RU

10.3 (± 1.3) RU

MPXV L1R-DS

1.3 RU

1.8 RU

1.1 RU

MPXV L1R-CSA

3.9 RU

2.7 RU

-0.7 RU

MPXV L1R-CSE

-1.2 RU

2.0 RU

3.1 RU

  1. How many repeats of the result Table 3, because this is no deviation data about KD value.

Response: As shown in the table notation, the data with (±) in parentheses represent standard deviations (SD) from the global fitting of five injections. The analysis was performed in triplicate and the SD of KD has now been added in our revised manuscript.

Table 2. Summary of kinetic data of MPXV A29 binding with heparin, DS, CSA and CSE *

ka (M-1s-1)

kd (s-1)

KD (M)

Heparin

27810 (± 280)

7.1 × 10-3 (± 5.0 × 10-5)

2.6 × 10-7 (± 1.7 × 10-8)

DS

4487 (± 54)

2.8 × 10-3 (± 2.6 × 10-5)

6.2 × 10-7 (± 1.7 × 10-9)

CSA

3730 (± 120)

3.2 × 10-3 (± 5.4 × 10-5)

8.4 × 10-7 (± 7.1 × 10-9)

CSE

4858 (± 72)

1.5 × 10-3 (± 4.7 × 10-5)

3.1 × 10-7 (± 6.1 × 10-9)

  1. In the 2.2 and 2.3 result part, the conclusion is not solid, especially the conclusion in 2.3, the RU does not mean the binding ability, only the ka or KD value can show the binding ability. Such as, in result part 2.3, the decrease of RU may be because PPS or MPS change the binding mode of A29 to heparin. The author can mix the PPS or MPS with A29, and then use the different concentrations to check the difference of ka or KD value by SPR.

Response: We agree with the reviewer’s comment and have changed “the binding ability” to “the binding signal” for solution competition SPR. The purpose of solution competition SPR is to test other GAGs or sulfated glycan for their ability to inhibit protein binding to heparin (see Figure 2).

Figure 2. Diagram of SPR solution competition study. The blue spheres represent TIMP-3 protein, the short red helices represent heparin or other GAGs. TIMP-3 protein (pre-mixed with 1000 nM heparin oligosaccharides or GAG) was injected over a chip containing immobilized heparin; only free protein can bind to the heparin on the chip. [S1]

[S1] Zhang, F.; Lee, K. B.; Linhardt, R. J., SPR Biosensor Probing the Interactions between TIMP-3 and Heparin/GAGs. Biosensors (Basel) 2015, 5, (3), 500-12.

  1. Does PPS or MPS have the unspecific binding with the Biotin labeled the heparin SA chips?

Response: No, PPS and MPS do not non-specifically bind the biotin on SA chips. The binding signal between PPS or MPS (1 µM) and biotinylated heparin are shown as below.

Figure S1. SPR sensorgrams of PPS and MPS binding with heparin chips. PPS or MPS (in 1000 nM) was injected to heparin chip, the control was MPXV A29 injected at 250 nM.

  1. What is the concentration of PPS or MPS pre-mix with A29, in the manuscript the author mentions the same concentration (Page 6, 209), but in the Fig.5 legend, it is “MPXV A29: 250 nM mixed with 1000 nM of sulfated glycans”.

Response: The same concentration (1000 nM) means the same concentration of all sulfated glycans, including heparin oligosaccharides, desulfated-heparins and different heparin analogs (PPS and MPS).  “MPXV A29 (in 250 nM) mixed with 1000 nM of sulfated glycans (heparin, PPS, and MPS)” is correct.

Reviewer 2 Report

Comments:

1.     Authors found that despite the difference in HBS sequence in VACV A27 and MPXV A29, both bind to heparin with similar affinity, indicating that the binding is sequence independent. Here it is notable that both these sequences differ only at two positions (K to N, and A to T), and it is possible that the binding is influenced by other residues (which remain similar) surrounding these variable residues. Using any alternative approach, like alpha-fold or docking, can the authors speculate what is the limiting factor for binding?

2.     The association kinetics of MPXV A29 with Heparin is fast (tight binding/more affinity) while with other GAGs it is slow. Authors suggest that this is because of higher degree of sulfation in heparin. Between CSE and DS, which one exhibit more degree of sulfation?

3.     In all the SPR experiments, how the phenomena of mass transfer taken care?

4.     Did Authors cross validated the observations found in this study using any other complimentary methods, like ITC or Thermal shift etc?

Author Response

Reviewer #2

  1. Authors found that despite the difference in HBS sequence in VACV A27 and MPXV A29, both bind to heparin with similar affinity, indicating that the binding is sequence independent. Here it is notable that both these sequences differ only at two positions (K to N, and A to T), and it is possible that the binding is influenced by other residues (which remain similar) surrounding these variable residues. Using any alternative approach, like alpha-fold or docking, can the authors speculate what is the limiting factor for binding?

Response: The finding that “despite the difference in HBS sequence in VACV A27 and MPXV A29, both bind to heparin with similar affinity” is quoted in the introduction of this paper (Ref [19] of the manuscript), but the work was not done by our group. In this paper, we focused on the interaction between MPA29 and GAGs, aiming at developing therapeutics and drugs for the treatment of MPXV. The suggestion proposed by the reviewer is a very promising project, and we now consider adding this part to in-depth experiments in future.

[19] Hughes, L. J.; Goldstein, J.; Pohl, J.; Hooper, J. W.; Lee Pitts, R.; Townsend, M. B.; Bagarozzi, D.; Damon, I. K.; Karem, K. L., A highly specific monoclonal antibody against monkeypox virus detects the heparin binding domain of A27. Virology 2014, 464-465, 264-273.

  1. The association kinetics of MPXV A29 with Heparin is fast (tight binding/more affinity) while with other GAGs it is slow. Authors suggest that this is because of higher degree of sulfation in heparin. Between CSE and DS, which one exhibit more degree of sulfation?

Response: CSE has a higher degree of sulfation than DS. CSE, chondroitin-4,6-sulfate, has two sulfate groups per disaccharide unit. DS is micro-heterogeneous, polydisperse. The primary structure of DS includes sulfation primarily at the 4-position of D-GalNAc and rarely at the 6-position of D-GalNAc and the 2-position of L-IdoA or D-GlcA residues. DS contains approximately one sulfate group per disaccharide unit. (Ref [25] of the manuscript)

[25] Linhardt, R. J.; Hileman, R. E., Dermatan sulfate as a potential therapeutic agent. General Pharmacology: The Vascular System 1995, 26, (3), 443-451.

  1. In all the SPR experiments, how the phenomena of mass transfer taken care?

Response: In SPR experiments, low responses are preferred over high responses. This is because high responses in general are affected by mass transport and non-1:1 interactions [S1, S2]. An appropriate level of immobilization is to immobilize enough ligand to get proper signals above the noise level, but keep the immobilization level low enough to avoid mass transport or steric hindrance. In addition, we used high flow rate (30 μl/ml) to reduce the possible effect of mass transfer.

[S2] O'Shannessy, D. J.; Winzor, D. J., Interpretation of deviations from pseudo-first-order kinetic behavior in the characterization of ligand binding by biosensor technology. Anal Biochem 1996, 236, (2), 275-83.

[S3] Schuck, P.; Minton, A. P., Analysis of mass transport-limited binding kinetics in evanescent wave biosensors. Anal Biochem 1996, 240, (2), 262-72.

  1. Did Authors cross validated the observations found in this study using any other complimentary methods, like ITC or Thermal shift etc?

Response: We did not use other complementary methods (such as ITC or Thermal shift) due to the limited amount of protein and sulfated glycans. We plan to perform NMR binding studies on A29-heparin in our future work.

Round 2

Reviewer 1 Report

The authors have addressed all of my concerns.